# The Late Pliocene jet stream: Changes and drivers of the mean state and variability

Abigail E. C. Buchan<sup>1</sup>, Alan M. Haywood<sup>1</sup>, Julia C. Tindall<sup>1</sup>, Stephen J. Hunter<sup>1</sup>, Aisling M. Dolan<sup>1</sup>, and Daniel J. Hill<sup>1</sup>

<sup>1</sup>School of Earth and Environment, University of Leeds, Woodhouse Lane, Leeds, West Yorkshire, LS2 9JT, UK **Correspondence:** Abigail E. C. Buchan (eeaecb@leeds.ac.uk)

Abstract. The Late Pliocene has frequently been used as a way to improve our understanding of the climate system in a warmer state. Larger scale features of Late Pliocene climate, such as Arctic Amplification, will impact global circulation including the jet stream. To date, the majority of Late Pliocene studies have focused on long term mean climate, however, considering interannual variability is important to fully understand the response of the climate system to different forcings. Using data from the Pliocene Model Intercomparison Project Phase 2, we find a more poleward, yet weaker jet stream in the North Pacific during winter months and increased interannual jet stream variability in the Late Pliocene compared to the pre-industrial control. This result is consistent across the majority of models, although there is variation in the magnitude of change across the ensemble. Using new simulations from the Hadley Centre Climate Model Version 3 (HadCM3), we find that changes in jet stream variability are due to orography and vegetation boundary conditions and are correlated with sea ice feedbacks. Carbon dioxide has little impact on the interannual variability in HadCM3. These differences in jet stream variability are linked to a decrease in meridional temperature gradient driven by an enhanced Atlantic Meridional Overturning Circulation. This is important as these differences might suggest a shift in the distributions of climate variables, such as temperature and precipitation, which could have implications for how proxy data and model simulations are compared. These changes in variability, and how the changes are represented in climate models, suggest the Pliocene is not an analogue for future jet stream interannual variability.

# 15 1 Introduction

20

# 1.1 Palaeoclimatology and Late Pliocene climate

As the global climate warms, the frequency and intensity of extreme events have increased, and events are occurring that are outwith the observational record (Robinson et al., 2021). Examining past climates can help the understanding of the behaviour and response of a warmer climate system to different forcings, such as changes to CO<sub>2</sub> and ice sheet configurations.

One time period well suited to this is the mid-Piacenzian Warm Period, specifically Marine Isotope Stage KM5c, 3.205 Ma (Haywood et al., 2013), hereafter referred to as the Late Pliocene. During this time, both orbital forcing and CO<sub>2</sub> levels (around 400 ppm) were similar to present day values (Haywood et al., 2016; Lan et al., 2025). Reconstructing CO<sub>2</sub> is difficult, so there is a large uncertainty on this value, with many strands of proxy evidence. From Boron isotope analysis, De La Vega et al.

30

55

(2020) report that CO<sub>2</sub> values during the KM5c interval to be  $391^{+30}_{-29}$  ppm with 95% confidence. A value of 400 ppm was chosen for some climate modelling studies to account for the forcing derived from other greenhouse gases that have no proxy reconstructions available (Haywood et al., 2016). Both the Greenland and the Antarctic ice sheets were reduced in size (Dolan et al., 2015; de Boer et al., 2015), leading to higher sea levels (Dowsett et al., 2016). Palaeogeography differs slightly from modern day, including the closure of northern hemisphere gateways such as the Bering Strait and the Canadian Archipelago (Dowsett et al., 2016).

Through the Pliocene Model Intercomparison Project (PlioMIP), Late Pliocene climate has been simulated by a number of climate models. PlioMIP phase 2 saw 17 climate models complete experiments to understand the behaviour of Late Pliocene climate, and the large-scale features of these simulations are described in Haywood et al. (2020), with the results of HadGEM3 presented separately in Williams et al. (2021). The ensemble multi model mean (MMM) temperature increase from preindustrial was 3.2°C with a global increase in precipitation of 7% (Haywood et al., 2020). Some other significant features of Late Pliocene climate include polar amplification (Haywood et al., 2020), a strengthened Atlantic Meridional Overturning Circulation (AMOC) (Zhang et al., 2021; Weiffenbach et al., 2023) and lower El-Niño variability (Oldeman et al., 2021; Pontes et al., 2022).

To date, the majority of studies focused on the Late Pliocene have looked at changes to the mean state. As the Late Pliocene gains momentum as a potential analogue for the future (Burke et al., 2018; Burton et al., 2025), and to improve our understanding of the climate during this time, it becomes important to consider higher-frequency variability. Considering variability and extremes within a palaeoclimate context could provide important considerations for the interpretation of both proxy and climate model data. In future climate projections, distributions of temperature and rainfall shift, with more extremes expected (Kodra and Ganguly, 2014; O'Gorman, 2015), it can be assumed that these distributions were also skewed, compared to pre-industrial, in past warmer climates like the Late Pliocene.

Obtaining well-dated proxy data at the temporal resolution needed to perform analysis on extreme events is challenging. It is therefore important to understand the drivers of extreme events and examine how they change, using climate models, before thinking about how they could occur in the proxy record. Furthermore, looking at extreme events in simulations of a warm climate, that has been compared with proxy data (McClymont et al., 2020; Tindall et al., 2022), can help us understand how climate models simulate extreme events.

## 50 1.2 Jet stream controls variability and its relationship to extremes

One key feature in the mid-latitudes that can impact extreme events is the jet stream. Jet streams are fast-flowing, narrow bands of air in the upper troposphere. In each hemisphere, there are two main jet streams, the polar jet which forms at the polar front, and the sub-tropical jet which forms at the upper boundary of the Hadley Cell. These jet streams form in response to temperature gradients and the Coriolis Force (Woollings, 2010).

As observations and climate projections show an amplified rate of warming in Arctic surface temperatures (Rantanen et al., 2022; McCrystall et al., 2021), the surface meridional temperature gradient is reduced. This would suggest an equator-ward shift in the jet stream. Despite this surface change, the jet has been observed to be shifting polewards and becoming more

60

wavy (Martin, 2021). Woollings et al. (2023) link this trend to a change in upper level temperature changes, with the upper troposphere warming more in the tropics than the polar regions.

Many other factors can impact jet stream behaviour both globally and regionally. For example, the topography of the Greenland Ice Sheet can influence the North Atlantic jet (White et al., 2019). Sea ice cover has also been linked to changes in mid-latitude circulation. Decreasing sea ice cover is linked to an equatorward shift in the jet stream in future projections, which could inhibit the poleward shift of the jet stream in future climate (Zappa et al., 2018; Screen et al., 2018). Feedbacks in the climate system from sea ice loss are also an important factor to consider when thinking about Arctic influence on the jet stream. Observational studies show links between sea ice loss and changes to cold season weather in the mid-latitudes (Cohen et al., 2014; Overland et al., 2015), although short observational records make it difficult to draw statistically signifiant conclusions from them. The same connection is debated in modelling studies with some studies finding no link (Cohen et al., 2019) and other studies finding weak, but significant links (Smith et al., 2022).

Shifts in the jet stream may lead to changes in the occurrence of extreme events in the mid-latitudes. Jet waviness has been linked to changes in extreme weather, although this is regionally dependent (Röthlisberger et al., 2016). This is a similar result to a study finding that jet stream tilt produces different regional responses over the European region (García-Burgos et al., 2023).

It is clear that no one change in the climate system is solely responsible for changes in jet stream behaviour, and is in response to a combination of forcings, which palaeoclimatology may be well placed to investigate.

#### 75 1.3 Jet stream in the Late Pliocene

To date, a few studies have investigated the behaviour of the jet stream during the Late Pliocene. Li et al. (2015) looked at mean state changes to the westerly winds across the mid-Pliocene (~3.3 to 3 Ma) with data from PlioMIP1 and found that the westerly winds were displaced poleward compared with the pre-industrial. A difference was noted in the response of the atmosphere only and coupled ocean-atmosphere models, with ocean-atmosphere models exhibiting a smaller polar shift compared to the atmosphere only models. This highlights the importance of ocean heat transport and sea ice dynamics on midlatitude circulation. As PlioMIP2 provided new boundary conditions, including the closure of Arctic Ocean gateways, leading to a better data-model comparison in the higher latitudes (Haywood et al., 2020), the PlioMIP2 ensemble will likely provide more insight into circulation changes in the Late Pliocene, relative to PlioMIP1.

Work has also been carried out looking at the drivers of jet stream change and variability. Using CCSM4-UoT it was noted that CO<sub>2</sub> had little impact on changes to the jet stream and that ice sheet and orography changes were more significant in the generation of a wave train (Menemenlis et al., 2021), although the forcing decomposition employed was not full as suggested by Lunt et al. (2021), which may neglect nonlinearities. Oldeman et al. (2024) also investigated jet stream variability in the North Pacific and found that in the Pliocene the jet stream was weaker and more variable. It was also found that non-CO<sub>2</sub> boundary conditions created the majority of the differences. Both of these studies only used one climate model in their analysis and may not capture the true change in the jet stream, as the representation of the jet stream varies between models due to model resolution and how drag is parameterised (Zappa et al., 2013; Pithan et al., 2016).

One challenge in using the LP for examining changes to jet stream behaviour is the lack of a direct proxy to perform data model comparison and ground climate simulations in the real world. One method of reconstructing past atmospheric circulation is through dust flux proxies. Abell et al. (2021) examined dust fluxes in the Pacific Ocean through the Pliocene into glacial periods and found that in the warmer Pliocene climate, the westerlies were poleward and weaker compared to Pleistocene glacial periods in the Pleistocene. This is in general agreement with simulated model results (Li et al., 2015).

This study aims to examine the causes of Late Pliocene jet stream changes, in the mean state and monthly variability, to understand more about the climate system from a multi-model perspective. This will also provide a perspective on the usefulness of the Late Pliocene as a future analogue for jet stream variability. To answer these questions, we use new and existing climate model simulations of the Late Pliocene. Details of the simulations and analysis techniques used are in section 2. Section 3 contains the results and an interpretation of the mean state jet, the variability in the jet stream and explores the roles of different forcings to the changes observed.

# 2 Methods

100

105

#### 2.1 Model simulations and data

#### 2.1.1 The PlioMIP2 ensemble

The PlioMIP2 ensemble consists of simulations completed by 17 coupled climate models. Each model contributes to the core experiments, a pre-industrial control (PI) simulation and a Late Pliocene (LP) simulation following the PlioMIP2 experimental design (Haywood et al., 2016). The PI and LP runs have 280ppm and 400ppm atmospheric CO<sub>2</sub> retrospectively, other greenhouse gasses are kept at the PI values in the LP simulations. The LP simulations also have reduced Greenland and Antarctic ice sheets, closed northern hemispheric ocean gateways (Canadian Archipelago and the Bering Strait), changes in land-sea mask in the Maritime Continent and changes to vegetation. A full description of the boundary conditions, and experimental design, can be found in Dowsett et al. (2016) and Haywood et al. (2016). The 15 PlioMIP2 models used to examine the mean state behaviour in this paper are found in Table 1. Two models in the PlioMIP2 ensemble, CESM2 and CCSM4\_Utrecht, are not examined here due to a difference in coordinate systems.

Here we follow the new naming convention adopted for PlioMIP Phase 3 (Haywood et al., 2024) to allow for consistency with future work and to improve the accessibility to those outwith the Pliocene climate modelling community. A comparison of the PlioMIP2 and 3 nomenclatures are found in Table 2.

#### 2.1.2 New model simulations using HadCM3

To disentangle the contributions of change in the jet stream from CO<sub>2</sub>, ice sheet and orography forcings, four new forcing factorisation experiments using the Hadley Centre Model Version 3 (HadCM3) were used. These simulations form part of the

**Table 1.** The PlioMIP2 models used in this study. The Climate Sensitivity (CS) and Earth System Sensitivity (ESS) are taken from Haywood et al. (2020) and Williams et al. (2021). Models with an asterisk (\*) have an unchanged land-sea mask in the LP simulations.

| Model ID     | Sponsor(s) and country            | Atmosphere<br>Resolution  | CS  | ESS | Reference                  |
|--------------|-----------------------------------|---------------------------|-----|-----|----------------------------|
| CCSM4        | National Center for Atmospheric   | 0.9 × 1.25                | 3.2 | 5.1 | Feng et al. (2020)         |
|              | Research (NCAR), USA              |                           |     |     |                            |
| CCSM4-UoT    | University of Toronto, Canada     | 0.9 × 1.25                | 3.2 | 7.3 | Chandan and Peltier (2017) |
| CESM1.2      | NCAR, USA                         | 0.9 × 1.25                | 4.1 | 7.7 | Feng et al. (2020)         |
| COSMOS       | Alfred Wegener Institute, Germany | $3.75 \times 3.75$        | 4.7 | 6.5 | Stepanek et al. (2020)     |
| EC-Earth3.3  | Stockholm University, Sweden      | 1.125 × 1.125             | 4.3 | 9.4 | Zheng et al. (2019)        |
| GISSE2.1G    | Goddard Institute for Space Stud- | 2.0 × 2.5                 | 3.3 | 4.0 | -                          |
|              | ies, USA                          |                           |     |     |                            |
| HadCM3       | University of Leeds, UK           | 2.5 × 3.75                | 3.5 | 5.6 | Hunter et al. (2019)       |
| HadGEM3*     | University of Bristol, UK         | 1.875 × 1.25              | 5.5 | 9.7 | Williams et al. (2021)     |
| IPSLCM6A     | Laboratoire des Sciences du Cli-  | 2.5 × 1.26                | 4.8 | 6.5 | Lurton et al. (2020)       |
|              | mat et de l'Environnement (LSCE), |                           |     |     |                            |
|              | France                            |                           |     |     |                            |
| IPSLCM5A2    | LSCE, France                      | 3.75 × 1.875              | 3.6 | 4.2 | Tan et al. (2020)          |
| IPSLCM5A     | LSCE, France                      | $3.75 \times 1.875$       | 4.1 | 4.5 | Tan et al. (2020)          |
| MIROC4m      | University of Tokyo, Japan        | $\sim 2.8 \times 2.8$     | 3.9 | 6.0 | Chan and Abe-Ouchi (2020)  |
| MRI-CGCM2.3* | University of Tsukuba, Japan      | $\sim 2.8 \times 2.8$     | 2.8 | 4.7 | Kamae et al. (2016)        |
| NorESM1-F    | Bjerknes Centre for Climate Re-   | 1.9 × 2.5                 | 2.3 | 3.3 | Li et al. (2015)           |
|              | search (BCCR), Norway             |                           |     |     |                            |
| NorESM-L     | BCCR, Norway                      | $\sim$ 3.75 $\times$ 3.75 | 3.1 | 4.1 | Li et al. (2015)           |

HadCM3 contribution to PlioMIP3. For a description of the model structure see Gordon et al. (2000) and updates in Valdes et al. (2017).

HadCM3, a coupled atmosphere-ocean general circulation model, has been extensively used for palaeoclimate studies including simulations of the Pliocene, for example Hunter et al. (2019). Due to a runtime of 50 to 100 model years per day, it can be run for long enough so that the climate can reach an equilibrium state. The atmosphere component has a horizontal resolution of 3.75° longitude by 2.5° latitude, with 19 vertical layers and a 30 minute time step. This makes it one of the lower resolution models within the PlioMIP2 ensemble (Table 1). However, it produces a good climate with LP warming of 2.9°C, close to the MMM (3.2°C) and within the model range (5.2-1.7°C) (Haywood et al., 2020). The model is run with prescribed static vegetation and uses the MOSES2.1 land surface scheme.

**Table 2.** Boundary conditions from a selection of PlioMIP3 runs. The PlioMIP2 nomenclature is also provided for ease of comparison to other studies.

| PlioMIP3 ID | PlioMIP2 ID        | CO <sub>2</sub> (ppm) | Orography | Ice sheets |
|-------------|--------------------|-----------------------|-----------|------------|
| PI          | $E^{280}$          | 280                   | PI        | PI         |
| LP          | Eoi <sup>400</sup> | 400                   | LP        | LP         |
| $LP^{280}$  | Eoi <sup>280</sup> | 280                   | LP        | LP         |
| $PI^{400}$  | $E^{400}$          | 400                   | PI        | PI         |
| PI_lp-orog  | $Eo^{280}$         | 280                   | LP        | PI         |
| LP_pi-ice   | $Eo^{400}$         | 400                   | LP        | PI         |
| PI_lp-ice   | Ei <sup>280</sup>  | 280                   | PI        | LP         |
| LP_pi-orog  | Ei <sup>400</sup>  | 400                   | PI        | LP         |

The ocean is a rigid-lid model with a horizontal resolution of 1.25° by 1.25° and 20 vertical unevenly spaced layers and a time step of 1 hour, which is coupled to the atmosphere at the end of each model day. Each atmospheric grid cell is associated with 6 ocean cells, meaning that the coastlines in the ocean surface layer have a resolution that matches the atmospheric component. For further information on the setup of HadCM3 for PlioMIP2 see Hunter et al. (2019).

To set up the forcings factorisation experiments (PI\_lp-orog, LP\_pi-ice, PI\_lp-ice, LP\_pi-orog), the PlioMIP3 experimental design in Haywood et al. (2024) was followed with details on the recommended set up in Haywood et al. (2016). It was decided to produce the full suite of forcing factorisation to be able to achieve a full forcing factorisation as described by Lunt et al. (2021). There is slight divergence from the original protocol, the enhanced LP boundary conditions were implemented opposed to the standard LP land sea mask. The experiments with LP orography and PI ice were started with a full LP run and the landmass of Greenland and everything south of 60° S was replaced with boundary conditions from the PI run. A similar approach is used for the experiments with PI orography and LP ice. The LP orography runs were spun off from an LP experiment with the same CO<sub>2</sub> value as the experiment and the PI orography runs were spun off from a PI experiment with the same CO<sub>2</sub> value. It is acknowledged that this does not fully separate out the ice sheets and orography as a change in the ice sheets, leads to a change in the orography and vegetation. The description of the boundary conditions used is in Table 2. The experiments were ran for 3500 model years to allow for them to reach an equilibrium state.

#### 5 2.1.3 The Lunt et al. (2021) factorisation method

These new simulations can be combined with existing experiments (PI, LP, PI<sup>400</sup> and LP<sup>280</sup>) to allow for a full forcing factorisation to be completed. To separate out the contribution of each of the factors listed in Table 2 on the LP climate we employ the linear-sum factorisation developed by Lunt et al. (2021):

$$\Delta x_1 = \frac{1}{6} \left[ 2(x_{100} - x_{000}) + (x_{110} - x_{010}) + (x_{101} - x_{001}) + 2(x_{111} - x_{011}) \right] \tag{1}$$

$$\Delta x_2 = \frac{1}{6} [2(x_{010} - x_{000}) + (x_{110} - x_{100}) + (x_{011} - x_{001}) + 2(x_{111} - x_{101})]$$
 (2)

$$\Delta x_3 = \frac{1}{6} \left[ 2(x_{001} - x_{000}) + (x_{101} - x_{100}) + (x_{011} - x_{010}) + 2(x_{111} - x_{110}) \right] \tag{3}$$

where x is the variable and the subscript represents the combination of  $CO_2$ , ice sheets, and orography used in that simulation. 1 represents LP conditions and 0 represents PI conditions. For example,  $x_{100}$  would be the experiment with LP  $CO_2$  but with PI orography and ice sheets, the  $PI^{400}$  experiment.  $x_{010}$  would represent the  $PI_p$ -ice experiment with PI  $CO_2$  and orography, but LP ice sheets. This factorisation is complete, unique, pure and symmetric (Lunt et al., 2021).

### 2.1.4 Reanalysis data

ERA5 reanalysis data is used to understand how well the pre-industrial control model simulations capture observations (Hersbach et al., 2023). ERA5 reanalysis data uses data assimilation from observations in combination with models to provide a record of global atmospheric conditions (Hersbach et al., 2020). ERA5 data spans from 1940 to present, producing a record shorter than the modelling runs presented here which may impact some of the variability metrics. Therefore, data from the NOAA–CIRES–DOE Twentieth Century Reanalysis version 3 (NOAA CR20) is also included here, as the record begins in 1806. NOAA CR20 only incorporates surface based observations which can lead to bias in the upper atmosphere (Slivinski et al., 2021). The reanalysis data will also contain influences of anthropogenic climate change, so is not a perfect comparison to model runs in climate equilibrium. Therefore, there is also potential for uncertainty in the comparison of the PlioMIP2 control runs to reanalysis data.

# 2.2 Analysis of model simulations

For the analysis, we take the final 100 years of each simulation and use the zonal wind speed and temperature fields at different pressure levels in the atmosphere. The model data was bi-linerally regridded to a 1° by 1° grid to allow for comparison between models and to construct a multi model mean.

Following the methods employed by Li et al. (2015), the latitude of the jet stream is defined as the latitude of maximum zonally averaged zonal wind, and the speed is taken as the maximum speed of the zonal wind. The latitude of maximum speed is found by performing a cubic spline interpolation around the points of maximum speed to achieve a latitude to 0.1°. As the jet stream and its impacts are regionally dependent, we study North Pacific and the North Atlantic separately. The North Atlantic region is defined as 15°N to 75°N, 60°W to 0 °E following Woollings et al. (2010). We define the North Pacific region as 15°N to 75°N, 160 °E to 220°E to be consistent with previous Pliocene studies of North Pacific jet stream variability (Oldeman et al., 2024). To understand the mechanisms of change we use the Lunt et al. (2021) method described in section 2.1.3 on the zonal wind speeds and upper level temperature change. We focus on Northern Hemisphere winter (December January and February

(DJF)) for this analysis as the jet stream is strongest during this time due to an increased meridional temperature gradient. Similar analysis could be repeated for the Southern Hemisphere, but is not done so here.

To measure the variability of the jet stream, we calculate the standard deviation of the latitude of maximum wind speed in the month of January. The results of the forcing factorisation variability are independent of the 100 year period of the simulation used.

# 3 Results and interpretation

# 3.1 Changes to the mean state jet stream

Unlike previous studies on the Late Pliocene jet stream that consider global zonal averages (Li et al., 2015), the behaviour of the zonal winds is studied separately for different regions. In the North Pacific region, the pre-industrial control MMM captures the reanalysis well (Fig 1. In the simulations, the zonal wind speeds are reduced in the Late Pliocene compared to the pre-industrial with a slight poleward shift seen in the MMM (Fig 1). The MMM poleward shift of the latitude of maximum zonal wind speed is 0.6°, 1.8° and 2.0° at 200, 500 and 850hPa respectively. This is in line with the expected response due to weakening on the meridional temperature gradient reported in the PlioMIP2 ensemble Haywood et al. (2020).

**Figure 1.** Multi-model mean zonal wind speeds for the PI (blue), LP (red) and LP-PI anomaly (orange). ERA5 reanalysis data is shown in the dashed black line. A-C is for the North Pacific region and D-F is for the North Atlantic. Shading represents the model range.

205

Although the MMM shifts, the range of model response cannot be ignored. As demonstrated in Fig 1 and 2, the models feature a large range of possible jet stream changes. Eight of the models agree with a poleward shift at 200 hPa and 13 of the 15 models agree with a poleward shift in at least one pressure level (Fig 2). Each model in the ensemble captures the general feature of the jet stream when comparing the pre-industrial control simulation against ERA5 reanalysis data, although some models are closer fits than others (Supplementary Fig 1).

**Figure 2.** Poleward shift of the latitude of maximum zonal wind in the North Pacific region in DJF at three pressure levels across all the models. Positive values indicated the jet is poleward in the Late Pliocene compared to the pre-industrial control simulations.

In the North Atlantic region, the presence of both the subtropical and polar jet is seen at 200hPa, with a double peak in the zonal wind profile (Fig 1D). None of the models, with the exception of HadGEM3, capture the reanalysis well at this level (Supplementary Fig 2), with the MMM being faster than the reanalysis. When considering the MMM change, a poleward shift, and a slight strengthening in the winds is observed. After removing the data points at 200hPa from CCSM4-UoT and CESM1.2 (due to them sampling either the polar or subtropical jet when changing between the Pre-industrial and the Late Pliocene), the North Atlantic jet stream shifts by 1.9, 0.4 and 1.1° at 200, 500 and 850 hPa respectively. There is also lower model spread with only two models exhibiting a large equatorward shift. The mean; however, is driven by some models with larger changes, such as HadCM3 and MRI-CGCM2.3 (Supplementary Fig 2).

Considering the spatial variations, the change is seen as a strengthening on the poleward flank and a weakening at the equatorward flank at both the 200 and 850hPa level (Fig 3). The MMM difference in Fig 3 is muted compared to the differences seen when examining the individual model differences. This may be due to differences in the placement of the jet stream and the magnitude of poleward shift cancelling in the MMM (Supplementary Fig 3 and 4).

In the North Pacific, the change seen at 850hPa has a similar spatial pattern in to the change at the 200hPa layer indicating that the zonal wind at each layer are linked which allows for comparison with proxy data. Abell et al. (2021) examine changes

220

225

240

in the strength and position of the jet stream using dust flux records in warmer climates compared to glacial climates and report that in warmer climates, like the Late Pliocene, the jet stream is poleward and weaker compared to cooler climates. This matches what is found in this modelling study. More work is needed to fully assess the dust flux records in different ocean basins to understand the full proxy signal. However, it is difficult to obtain data at a high enough temporal resolution to support this analysis. It is also possible that changes in dust flux, and aerosols may lead to changes in atmospheric circulation if there is a change in land surface type although Abell et al. (2021) argue that atmospheric circulation is the dominant driving factor of the observed flux changes.

There are several reasons for differences between models. First, is the resolution of the model, with models with a finer spatial resolution able to capture more complex ocean-atmosphere interactions, which play a large role in the behaviour of the jet stream. Some models are obvious outliers from the reanalysis data, such as IPSLCM5A, IPSLCM5A2, COSMOS, HadCM3 and NorESM-L (Supplementary Fig 1 and 2). These are the models among the lowest spatial resolution of PlioMIP2 (Table 1). This indicates that model resolution is an important factor to consider when examining change in the jet stream. There is also added uncertainty in the representation of the jet stream within the ERA5 reanalysis, as observations higher in the atmosphere are sparse. Some models also better capture proxy reconstructions of sea surface temperatures, for example, CCSM4-UoT, CESM1.2, IPSLCM6A and MIROCm4m (Haywood et al., 2020). These models may also be able to provide a better representation of jet stream changes during the Pliocene.

To examine a possible causes of the changes to the mean state jet, the change in jets stream latitude is compared to the ratio of the earth system sensitivity (ESS) and the equilibrium climate sensitivity (ECS) (Supplementary Fig 5). However, no strong or significant relationship was detected suggesting that drivers of the change in the jet may not be consistent across the range of models. To fully understand the drivers of jet stream change, each model will need to be individually examined.

#### 230 3.2 Controls on the mean state shift using HadCM3

To understand which of the Pliocene boundary conditions contribute to the change in the mean state jet stream we use the HadCM3 forcing factorisation experiments described in section 2.1.2. The surface temperature forcing decomposition can be seen in Supplementary Fig 6. This highlights the regional influences of changes to temperature from ice sheets and orography in the areas of largest change. Fig 4 shows the total change in zonal wind speed between the PI and the LP and the contribution of CO<sub>2</sub>, ice sheets and orography. The change in wind speed due to CO<sub>2</sub> is small, with the largest contribution being orography (including land-sea mask and vegetation) and ice sheets following as the second leading cause in the change. These changes are related to the upper level temperature gradients as seen in Fig 5. CO<sub>2</sub> causes a cooling in the upper polar regions, and warming in the upper tropical regions which is consistent with observed patterns under modern climate change (Ladstädter et al., 2023). The temperature changes due to changes in the orography also reveal a large change in the meridional temperature gradient.

As the jet stream forms in response to temperature gradients, a weaker temperature gradient may lead to a weaker and poleward jet stream. Examining the behaviour of the AMOC in each experiment (Fig 6), the AMOC in the LP is stronger than in the PI, this is consistent with other models in the PlioMIP2 ensemble (Weiffenbach et al., 2023). The experiments employing LP orography show a stronger AMOC than the experiments with PI orography. This is likely due to changes in

250

**Figure 3.** Multi-model mean zonal wind speeds for 200hPa (left) and 850hPa (right). The lower plots show the difference between the Late Pliocene and the Pre-Industrial. Note the difference in scales between the two pressure levels.

the northern hemisphere ocean gateways. This is consistent with previous studies, which relate the increase in LP AMOC to the closure of the Arctic Ocean gateways (Weiffenbach et al., 2023). This enhanced AMOC due to orography changes is one contributor to the changes in temperature gradient, as more heat is transported northward. A change in vegetation can also lead to a large change in temperature, due to different plant types having different impacts on temperature through albedo and evapotranspiration (Bonfils et al., 2012). It is likely that the changes seen here are a combination of ocean circulation and vegetation changes. Further investigation could include vegetation changes in the suite of forcing fractionation experiments, in order to fully understand how it contributes to the change in the jet stream. This would require a set of 16 simulations (Lunt et al., 2021).

It should also be noted that it is impossible to fully disentangle the contribution of ice sheets and orography because the two are inherently interlinked. The ice sheet itself provides a change in orography, which is especially important for controlling

changes in the North Atlantic jet stream with the position of Greenland being important for the pattern of the jet stream in that region (White et al., 2019).

**Figure 4.** Change in 200hPa zonal wind speed between the Late Pliocene and the Pre-industrial in HadCM3 and the contribution of the change from CO<sub>2</sub>, ice sheet and orography forcings to the total change.

**Figure 5.** Change in 200hPa temperature between the Late Pliocene and the Pre-industrial in HadCM3 and the contribution of the change from CO<sub>2</sub>, ice sheet and orography forcings to the total change.

**Figure 6.** AMOC in the HadCM3 for the PI and LP and the four forcing factorisation experiments. Values show the maximum value in each experiment.

#### 3.3 Jet stream variability

260

265

Although jet stream variability has been looked at in similar manner before within CCSM4-Utrecht (Oldeman et al., 2024), as discussed in section 3.2, there are large model differences in the mean state shift and this will translate to model differences in variability. To assess the model performance against ERA5 reanalysis data we compare the mean latitude and variability in position the the PI simulations against reanalysis data (Fig 7). The majority of models capture the the variability but there is a larger spread in the mean latitude. The impact of multidecadal variability cannot be ignored in this assessment and some longer modes of variability may be influencing the 100-year mean.

The response to Late Pliocene boundary conditions across models is varied (Fig 8). The model-to-model variability is larger than the PI to LP change, suggesting that no conclusions can be drawn from this data set. However, some models may be better at capturing the change in the jet stream than others. For example, HadGEM3 and MRI-CGCM2.4 have an unchanged land–sea mask. Since we expect the jet changes to be due to orography changes, we do not expect a large change in the jet stream in these models, as is seen in Fig 8.

**Figure 7.** Latitude of maximum zonal wind speed and standard deviation of this latitude in the PlioMIP2 PI simulations compared to the ERA5 reanalysis data (dashed gray lines).

The variability between the two time periods is increased in most models. However, a lot of this change is only small (Fig 9). HadCM3, CCSM4-UoT and MIROC4m all show a larger change in the variability of the jet stream. Two models that show an increased speed and a decreased variability, COSMOS and NorESM-L are the two models with the lowest resolution out of the PlioMIP2 ensemble reinforcing that resolution does matter when considering the jet stream. MRI-CGCM2.3 also shows an increase in speed and decrease in variability which could be related to the unchanged land-sea mask in the LP experiment in this model. The models with a LP land-sea mask and higher spacial. resolution, indicate that the jet steam is weaker and more variable.

**Figure 8.** Hovmöller diagrams for the zonal mean wind speed at 200hPa for in January for 100 years for the North Pacific. The left-hand side is from the PI experiments and the right-hand side is from the LP experiments with the exception of the bottom two panels which show the reanalysis data. The dashed black line indicates the latitude of maximum zonal wind speed.

285

**Figure 9.** Change in the mean speed of the jet steam and the change in the variability of the jet stream position. Each model in the ensemble is labelled. The dashed gray lines divide the graph into quadrants where the upper left section shows models with an increased speed and decreased variability, and the lower right quadrant contains models with a slower speed and increased variability.

#### 275 3.3.1 Forcing decomposition of jet stream variability

In HadCM3, the North Pacific LP jet stream is weaker and more variable than in the PI (Fig 9) in agreement with the majority of the models. Using the new forcing factorisation experiments the contribution of each boundary condition change can be assessed. Figure 10 shows the jet stream variability in each of the 8 runs. There is little contribution of CO<sub>2</sub> to the total change, evidenced by experiments with the same ice sheets and orography, but a change in CO<sub>2</sub> having no visible change in speed or variability. The largest change comes from the alterations to the orography boundary conditions. This relates to the change in the mean state with the orography having the largest impact on the zonal wind speeds (Fig 4), likely due to orography having the largest impact on the meridional temperature gradient (Fig 5). As the largest changes in variability are due to non-CO<sub>2</sub> boundary conditions, this suggests the LP is not an analogue for future, CO<sub>2</sub> driven, jet stream variability. Despite this, the reduction in meridional temperature gradient leading to c achange in the jet stream could be application to future climate. The results of the forcing factorisation hold, independent of the 100 year period chosen to perform this analysis.

As declining sea ice has been shown to have links with changes in mid-latitude winter time circulation, the relationship between sea ice area and some jet stream metrics have been included here. Figure 11 shows a clear grouping of the experiments using LP orography vs experiments with PI orography. The split in the sea ice could be explained by either a stronger AMOC or a reduction in Arctic sea surface area due to the implementation of LP boundary conditions (or a combination of both). Sea ice area feedbacks positively on Arctic amplification, with a reduction in sea ice cover lowering albedo, which in turn warms

**Figure 10.** Hovmöller diagrams for the zonal mean wind speed at 200hPa for 100 Januaries for the North Pacific. The dashed black line indicates the latitude of maximum zonal wind speed.

**Figure 11.** January northern hemisphere sea ice area and jet stream speed, and jet latitude variability in HadCM3. The red box groups experiments with Late Pliocene orography and the blue box groups experiments with pre-industrial orography.

the Arctic (Jenkins and Dai, 2021). This Arctic amplification, leading to a weakening in the meridional temperature gradient, could create a slower and more wavy jet stream (Francis and Vavrus, 2015).

The CCSM4-UoT and COSMOS models also provided forcing factorisation experiments. The variability in the jet stream across the 8 simulations for these models can be found in Supplementary Figs 7 and 8. The COSMOS model is an outlier

as the jet stream was stronger and less variable in the LP, although it is unclear from the forcing factorisation what causes this. There is little change between all of the experiments with the exception of a strengthening in the full LP experiment, potentially a feature of a lower resolution model. In CCSM4-UoT CO2 has little impact on the speed and variability of the jet but both orography and ice sheets contribute to the slowdown and increased variability observed in the LP experiment. This equal contribution from ice sheets and orography was also noted in previous studies that examined upper troposphere dynamics (Menemenlis et al., 2021). Sea ice extent was also examined in CCSM4-UoT (Supplementary Fig 9). Here, a larger 300 contribution from ice sheet changes to variability and speed is also seen. In CCSM4-UoT, there is no significant difference in the AMOC amongst the forcing factorisation boundary experiments (Chandan and Peltier, 2018), meaning that northward heat transport may not be different between the runs. This could explain why the jet CCSM4-UoT is more sensitive to ice sheet changes than in HadCM3. Part of this could be due to the boundary conditions being slightly different in each model, 305 with the land sea mask around Antarctica being treated differently in each model which may create some of the difference observed in the AMOC. The model dependency of this variability change highlights the need for variability to be studied from a multi-model perspective, especially in a palaeoclimate setting where there is often only one realisation of each experimental set up limiting the ability to assess internal variability of climate models.

# 4 Conclusions and future directions

By examining models within the PlioMIP2 ensemble, it is found that during the LP, the jet stream was weaker and exhibits a poleward shift in the North Pacific region compared to the PI control period. This varies between models due to a number of factors including model resolution and extent of land sea mask modification. Examining higher frequency variability shows that the jet stream may be more variable in the LP, although again this is model dependent. However, the models with the largest reduction in jet stream speed also show the largest increase in variability. From new experiments using HadCM3, it is found that a change in the orography, including land–sea mask and vegetation, is the leading cause of the change in the jet stream due the impact on the meridional temperature gradient from AMOC, vegetation and sea ice changes. This could also explain why models that do not change the land-sea mask do not see a weaker or more variable jet in the LP.

As we approach a warmer, unknown future, studies have invoked the Late Pliocene as an analogue for future climate given the similarity of global mean Pliocene temperatures to projections for the end of this century (Burke et al., 2018; Burton et al., 2023). As the majority of the difference in the jet stream variability is caused by changes to orography, this element of the LP climate is not relevant to future change. Although the mechanism of change (a reduced temperature gradient) may be similar to what is projected for the future, as the direct causes of this are different from the future, comparisons need to be made with care. Impacts of the reduced ice sheets may be relevant for longer time scales as a reduction in the ice sheet size occurs, but as the climate is a complex system with many interactions and feedbacks, it may not be possible to apply this result directly to future projections. This agrees with previous studies on jet stream variability, which states that the Late Pliocene is not an analogue for future Northen Hemisphere winter time variability (Oldeman et al., 2021).

This work could however have impacts on the study of the Late Pliocene climate. As the jet stream becomes weaker and more variable, more persistent weather patterns may occur leading to a change in the frequency and intensity of extreme events. A change in the distribution of key climate variables could impact the way that model comparison with proxy data is interpreted. As higher frequency temporal variability is lost when performing proxy analysis, a change in variability between the reference period and the time period in question could change the way that the climate signal is transferred into a proxy variable, e.g. temperature. Further work is needed to fully understand how this change in the jet stream relates to surface changes, for example by studying how the jet stream interacts with different modes of variability.

This paper shows the use of considering interannual climate variability, and the drivers of it, for studies of the Late Pliocene,
particularly when thinking of it as an analogue for future climate. As there is range of responses across models, this work also
highlights the need for multi-model comparisons for assessing a change in internal variability.

Author contributions. AECB led the study, produced the model runs, conducted the formal analysis of the data and wrote the initial draft of the paper. AECB, AMH, JCT, AMD and DJH discussed the analysis and interpretation of the data. JCT and SJH supported the production of the modelling runs. All authors contributed to the preparation of the paper.

Competing interests. The authors declare that they have no conflict of interest.

Acknowledgements. AECB acknowledges that this work was supported by the Leeds-York-Hull Natural Environment Research Council (NERC) Doctoral Training Partnership (DTP) Panorama under grant no. NE/S007458/1.

This work was undertaken on ARC4, part of the High Performance Computing facilities at the University of Leeds, UK.

Support for the Twentieth Century Reanalysis Project version 3 dataset is provided by the U.S. Department of Energy, Office of Science Biological and Environmental Research (BER), by the National Oceanic and Atmospheric Administration Climate Program Office, and by the NOAA Physical Sciences Laboratory.

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
