# Peer review of "The Late Pliocene jet stream: Changes and drivers of the mean state and variability"

_EGUsphere, 2025_

## Referee Comment (RC1)

Comments on *The Late Plioene jet stream: Changes and drivers of the mean state and variability* by Buchan et al.

This paper is an interesting and relevant study of changes in jet stream behaviour between simulations of the Plioene and Pre-Industrial reference. It uncludes both a multi-model assessment of the PlioMIP2 ensemble and new sensitivity studies using HadCM3, adding to its novelty and robustness. The results clearly show that care must be taken when considering relevant dynamics of the atmosphere, as well as their variability, when considering the Pliocene as a potential future analog.

While the paper is generally relevant and the analyses seem sound, there is considerable ambiguity regarding the physical background of jet stream dynamics. This includes many subtleties which could fundamentally change the interpretation of some of the results. In my opinion, more care should be taken into clearly stating how the jet stream is defined, which different definitions can be used, how this translates to the analyses done as well as the results shown, and perhaps most importantly how this connects to the existing literature and overall motivation of this study. Improving the text in this regard would thus greatly improve the overall quality, interpretability and importance of this study.

Please do not consider my extensive comments as pure criticism, but just as much as an expression of my general interest into reading your work.

**General comments:**

Introduction: the paper needs a clearer statement on what is considered as 'the jet stream', from the context, I assume this is the polar/eddy-driven jet near the tropopause. In monthly mean zonal averages of zonal winds, the subtropical jet is usually much more prominent compared to the eddy-driven jet owing to the large difference in meridional variation. To clearly distinguish the eddy-driven jet in model output, one would either have to use sub-daily frequency or monthly means of the eddy fields (i.e. time-mean of U*U and V*V).
In addition, there is no clear consensus in literature on the height/pressure level to study the jet stream, Many studies use e.g. 850 or 500 hPa levels, or the vertical maximum within any possible range between 900-100hPa. A clear example of this is the Abell et al. study mentioned on L94, in which the dust proxy is a clear indicator of low level westerlies (surface to ~850hPa), but much less intuitive regarding the tropopause jet stream. Without further clarity on these possibilities, it is difficult to adequately compare the conclusions made between different studies.
Focussing on the Pacific and Atlantic basins does improve the ability to detect jet stream maxima in a zonal average sense. This is shown by the double het maximum over the Atlantic. As shown in Oldeman 2024, this double jet max can be related to a pattern of persistent anticyclonic wave braking over the North Atlantic Ocean (also shown in figure 3 of this manuscript). This is a known phenomenon in PI/PD conditions, which should be mentioned up front, as it is relevant to interpret shifts in jet latitude and strength.

From the introduction/methods section it is a bit unclear to me what the focus of this paper is. There is mentioning of earlier studies focusing on a single model, rightfully stating this as a main limitation. Further down, most of the focus in the methods is on new single-model experiments which seems a bit inconsistent. While part of the results are still on the ensemble, which are then complemented with single model experiments, I feel that this is not stated clearly enough early on. My suggestion is not to change the overall setup, but to slightly alter the focus and/or the related communication.

In the methods section, it is stated that the analysis follows that of Li et al 2015 who consider winds at 850, 500 and 250hPa (edit: this is specified in the results section, maybe mention in the methods?). Please clarify what is studied here. In addition, taking a single maximum from the zonal average leaves out all of the zonal variability in jet latitude and strength within a single time frame. This is a choice that may be justified, but please be more clear and motivate why. As shown in Oldeman et al. 2024, double jet maxima may considerably complicate the analysis and interpretation of jet strength. In addition, there is no clear statement on which time means are considered. The study considers monthly data, but are these averaged over the winter season (DJF) for each year? For variability, there is a brief mentioning of January means, are these then different from the time-mean analyses?

Subsection 3.2 in the results needs a lot more care regarding the 3D structure of subtropical and eddy-driven jet streams, as well as how this temporally varying structure is represented in the time means at a single pressure level. The link between temperature gradients and wind magnitude is implied but mostly lacks a proper explanation or background. These results are still relevant and the analyses seem sound, but considerably more care should go into motivating as well as interpreting them.

Subsection 3.3.1 needs some better structure; it quickly jumps between a large number of figures and generally lacks a solid motivation, interpretation and above all connection between the different results mentioned. This makes it tough to see the general picture and main message here.

**Specific comments**

L11 'This is important as …' Could the statement be further clarified or specified by reworking the sentence to make it a bit less vague?

L58 At upper levels, there is indeed an enhanced meridional temperature gradient in a warmer world (particularly through greenhouse gas-induced warming), but this mainly results from lower stratospheric cooling at higher latitudes as opposed to upper tropospheric warming at lower latitudes. Both can be at the same height/pressure level, owing to the meridional structure of the tropopause. Please clarify this.

L113 CESM2 and CCSM4_Utr are not used due to a difference in coordinate system; please explain? The latter was used to study the jet stream in Oldeman et al. 2024, so apart from being possible, including this into the study would help compare the results.

L115 Using the PlioMIP3 nomenclature does not seem intuitive to me, as this study considers PliMIP2 model output. Would it make sense to list the PlioMIP3 nomenclature instead and use the PlioMIP2 one in this paper to make it more comparable to previous work?

L128 please specify 'a good climate'

L130 I was puzzled by the statement on the vertical levels for a moment, until I noticed I missed the 'un' in unevenly spaced levels. Maybe rephrase slightly for clarity?

L158 ERA5 consists of a single model (i.e. IFS), rather than models? In addition, the 85 years spanned by ERA5 could be considered as similar to the 100 years in the PlioMIP ensemble? Using different reanalysis datasets is always helpful for a more complete comparison, but it is at least as important to consider its reliability (in addition to reduced observation methods/counts) in the pre-WWII period.

L170 I completely miss a statement on which vertical level or pressure level is considered to determine the jet stream.

L196 I am uncomfortable with the use of subtropical/polar jet here. As shown in figure 3, the jet stream pattern over the North Atlantic can be linked to persistent anticyclonic wave breaking. This causes the eddy-driven component of the jet to be dominant over the western/central part of the basin and the 'conventional' subtropical jet to regain strength over the eastern part. As you are considering time means of zonal wind at 200hPa, the analysis is strongly biased towards showing the subtropical jet. The pattern over the Atlantic is an exception to this rule that deserves much more care and attention interpreting the results.

L208 Linking the 850hPa and 200hPa levels is indeed useful for proxy comparisons, but doing this based on a qualitative comparison between both fields in the MMM is not very robust. A correlation between different models and/or years would provide a much stronger argument.

L220 HadCM3 is noted as a clear outlier when looking at the reanalysis data, this strongly advocates some further discussion on the interpretation of the model-specific analyses further down.

L235 Apart from the CO2 response being weaker, it is also opposite in sign compared to the other 2 forcings.

L236 I assume looking at the temperature responses is motivated by thermal wind balance, but I do not see this being mentioned? In that case, looking at the temperature response integrated over the atmospheric layer below would be more suitable. Furthermore, comparing the meridional temperature gradient response to CO2 and ice

sheets versus the wind speed response, seems to be rather inconsistent. This discrepancy (if correct) in the results seems to be missed out on or ignored altogether.

L237 Please be more specific regarding 'upper polar/upper tropical' regions, as this may imply anything from the top of the boundary layer to the top of the atmosphere.

L240 A weaker temperature gradient may lead to a weaker and poleward jet stream; what is the latter statement based on? Does this hold for the jet near the tropopause, or is it only valid at 850hPa? The first argument seems to be the complete opposite of what is shown in the figures, with generally enhanced temperature gradients at upper levels. Furthermore, over the Atlantic Ocean, there is no clear change in strength or average latitude, as the breaking wave pattern is reduced in strength and converges towards a single jet latitude in LP compared to PI. In addition to a more general shift, the opposite is seen over the North Pacific, both of these responses are consistent with Oldeman 2024.

L245 Also cite the Otto-Bliesner 2017 paper here?

L257 This sentence is rather tough to understand; are you talking about model differences in general, or specifically between CCSM4 and HADCM3? Why would a difference in the mean automatically imply the same for variability? (I'm not saying it does not, I am just unsure why).

L259 Does this consider the full NH, as opposed to Atlantic/Pacifc before? What would be the reason to differ from the previous analyses? Considering the large differences between the basins certainly limits the interpretation of these results.
Edit: I see the figure caption mentions North Pacific, please clarify in the text?

L276 There needs to be a clear explanation of what is considered as 'jet stream' variability, as the link between strength and variability is made multiple times in this work (and shown to be significant in Figure 9). There is, however, a substantial difference between spatial variations (i.e. 'wavy' jets) and the temporal variation of the position of the maximum in zonal average zonal wind speed. Both may be related, but this would need some proper motivation.

L292 I have seen the suggestion of linking a weaker meridional temperature gradient to a weaker and more wavy jet stream before, but there does not seem to be a clear physical mechanism nor observational evidence for this? Note that, regarding the above statement, wavy jets are not the same as temporally changing latitudes of the zonal wind max. Making any claims on wavy jets would require a much more detailed analysis of spatial patterns at high temporal frequencies and/or eddy components of velocities and fluxes.

**Figures:**

Figures
In general, please be more consistent with the sizes of e.g. fonts and colourbars between the different figures.
Also: add lat,lon coordinates to the spatial figures?

Figure 1 Please add a vertical dashed line or grid line showing the zero value to interpret that LP-PI change. Also consider scaling the change (e.g. x10) for readability.
Minor suggestion: while I appreciate the consistent scaling, the range in wind speeds can be reduced considerably for the 500/850hPa panels, improving readability. If consistent scaling is desired, you may adjust the panel width accordingly as well.
Is the figure showing DJF, January, or annual mean?

Figure 3 Please make use of a diverging colourmap, or a shift in colour for values that are below zero i.e. showing easterly winds. Again, also indicate whether this is showing boreal winter, winter in general, or something else?
The scaling of the difference plots could also be reduced to improve clarity?

Figure 4 Adding just a single contour for the PI reference (e.g. 30m/s) would really help interpret whether the changes mean a change in strength or a spatial shift in these panels.

Figure 6 The colourbar in combination with the contour lines is pretty rough. In addition, the relevance of this figure in the main text seems limited, as this is only used to argue that the AMOC is indeed stronger in the LP versus PI experiments, being consistent with previous work? The full 2D structure of the overturning stream function is of limited added value to this study.

Figure 8 While this is a rather intense multi-panel figure, I do appreciate the complete overview among models. For comparison, it could be helpful to add the numbers of mean and variability for each case, which are otherwise not shown?

**Errors/typos**

L28 Northern Hemisphere? (also on L244)
L30 was simulated?
L73 not one?
L92 LP stands for Late Pliocene?
L168 by-linearly?
L190 use \citep?
L208 in comparison to the change?
L217 slightly akward sentence, maybe rephrase?
L226 a possible causes, jets stream
L270 redundant period?
Fig8 caption for in January
L283 CO2-driven?
L284 to c achange ... could be application to
L290 feedbacks positively
L326 Oldeman 2024?